# Correlation between Malocclusion and Allergic Rhinitis in Pediatric Patients: A Systematic Review

**DOI:** 10.3390/children7120260

**Published:** 2020-11-27

**Authors:** Marco Farronato, Valentina Lanteri, Andrea Fama, Cinzia Maspero

**Affiliations:** 1Department of Biomedical, Surgical and Dental Sciences, School of Dentistry, University of Milan, 20100 Milan, Italy; marco.farronato@unimi.it (M.F.); valentinalateri78@gmail.com (V.L.); andreas.fama@hotmail.it (A.F.); 2Fondazione IRCCS Cà Granda, Ospedale Maggiore Policlinico, 20100 Milan, Italy

**Keywords:** allergic, malocclusion/pathology, mouth breathing/pathology, rhinitis

## Abstract

Background: Oral breathing, nasal obstruction and airway space reduction are usually reported as associated to allergic rhinitis. They have been linked to altered facial patterns and dento-skeletal changes. However, no firm correlation based on the evidence has been established. This systematic review has been undertaken to evaluate the available evidence between malocclusion and allergic rhinitis in pediatric patients. Methods: The research refers to Preferred Reporting Items for Systematic Reviews and Meta-Analyses Protocols (PRISMA-P) guidelines, databases (Medline, Cochrane Library, Pubmed, Embase and Google Scholar) were screened, the quality was evaluated through Quality Assessment of Diagnosfic Accuracy Studies (QUADAS-2). Results: The articles selected (6 out of initial 1782) were divided on the basis of the study design: two observational randomized study, three case–control study, one descriptive cross-sectional study, and one longitudinal study. A total of 2188 patients were considered. Different results were reported as related to allergic rhinitis ranging from a higher incidence of dental malocclusion, to an increase of palatal depth, and in posterior cross-bite about anterior open-bite and to longer faces and shorter maxillas. Conclusions: Most of the studies selected found a rise in the prevalence of both malocclusion and allergic rhinitis in children. However, the level of bias is high, impaired by a poor design and no conclusive evidence can be drawn.

## 1. Introduction

Allergic rhinitis (AR) is a chronic disease affecting children [1,2]. AR is often associated with asthma, sinusitis, conjunctivitis, hypertrophy of the lymphoid tissue and obstructive sleep apnea (OSAS), and it is seldom detected as an isolated pathology [3,4,5,6,7,8,9,10].

The prevalence of asthma in pediatric patients with a diagnosis of allergic rhinitis ranges between 75% and 80% [6,11].

Several studies have described that these patients also display mouth breathing. Oral breathing refers to the pathological condition in which the airflow during the breathing cycle at rest mostly runs through the oropharyngeal canal rather than the nasopharyngeal one. Oral breathing has been identified as being responsible for determining systemic changes that may cause morphological alterations at the craniofacial level [12,13,14]. An oral respiratory pattern may result from a partial obstruction of the nasal cavities, such as adenoid and tonsil hypertrophy, with increased air resistance, or alteration of the growth of dentoskeletal structures and the resulting modification of mandibular posture [15,16,17].

The passage from nasal to oral respiration can determine altered development of the dentoskeletal structures and of the nasal capsule, resulting in an anatomical-functional adaptation of the neuromuscular system, having important repercussions on craniofacial morphology. The low rest postural position of the tongue, indispensable in allowing the intraoral airflow and associated with increased pressure induced by the cheeks due to the augmented vertical dimension, can cause a transversal deficit in the upper jaw [18,19,20,21,22].

Moreover, a lowered jaw position would cause an increase in the vertical dimension due to the supereruption of the posterior teeth and the posterior rotation of the mandible. These characteristics are associated with the development of anomalies in craniofacial structures such as adenoid facies, also called as the “long face syndrome”, which refers to a long, open-mouthed face with short upper lip, prominent incisors, high arched palate and elevated nostrils. At the skeletal level, the face is narrow and elongated, with a tendency to bimaxillary retrognathism, to hyperdivergence, to a decreased nasopharyngeal airway depth and to the transverse hypoplasia of the maxilla [23,24,25,26,27,28,29,30,31].

It is frequently associated with atypical deglutition situations and speech disorders [32,33,34,35,36].

Other authors have ascribed the expression of these facial features to heredity, suggesting that mouth breathing may be an aggravating factor in patients with a dolichofacial pattern [37,38,39,40].

The aim of this systematic review is to assess whether there is evidence regarding a correlation between malocclusion and allergic rhinitis in children. A systematic literature review was performed, incorporating studies on children and adolescents assessing the prevalence of rhinitis in children with malocclusion and the prevalence of malocclusion in children with rhinitis.

## 2. Materials and Methods

This review is reported according to the PRISMA guidelines 2015 [41].

### 2.1. Protocol and Registration

The protocol for this systematic review was written before commencing the study and was developed in accordance with the PRISMA guidelines and the guidelines of the University of Milan for systematic reviews of the literature. After full consensus among the authors was achieved, the protocol was registered in PROSPERO with RN: CRD42019117546.

### 2.2. Search Strategy

The review was conducted through the following electronic databases: Medline, Pubmed, Embase and Cochrane Library, Google Scholar. The mesh terms used were ((“rhinitis, allergic” [MeSH Terms] or (“rhinitis” [All Fields] and “allergic” [All Fields]) or “allergic rhinitis” [All Fields] or (“allergic” [All Fields] and “rhinitis” [All Fields])) and (“malocclusion” [ Medical Subject Headings MeSH Terms] or “malocclusion” [All Fields]) and (“rhinitis, allergic, seasonal” [MeSH Terms] or (“rhinitis” [All Fields] and “allergic” [All Fields] and “seasonal” [All Fields]) or “seasonal allergic rhinitis” [All Fields] or “pollinosis” [All Fields]). The mesh terms refer to the Pubmed database.

### 2.3. Inclusion Criteria

We considered only articles with a reliable or accurate diagnostic method, information about etiology, diagnosis and therapy during pediatric ages, and articles specifically related to the relationship between allergic rhinitis and malocclusion.

Using the “limits” option, the authors considered only articles referring to “Humans”. Articles published up until August 2020 were considered. The following types of studies were included: observational studies, longitudinal studies, prospective studies, case–control studies, systematic reviews and clinical trials. Articles in languages other than English, but with English abstract were considered. Commentaries, letters to the editor and short communications were not considered. Other types of rhinitis were not considered, including non-allergic, infectious, medicamentosa, atrophic, sicca, and polypous rhinitis. Additionally, data related to adults were excluded. The inclusion and exclusion criteria are summarized in Table 1.

### 2.4. Quality Assessment

The Quality Assessment of Diagnostic Accuracy Studies tool-2 (QUADAS-2) was used to evaluate the risk of bias [42]. The reviewers assessed the risk of bias of each study independently, and discrepancies were resolved by a third reviewer.

### 2.5. Data Extraction

Two reviewers were responsible for the selection of articles, without blinding to the authors. Disagreements were discussed among the authors until agreement was achieved. The data extracted included: journal and year of publication, study design, therapeutic outcomes and the author’s conclusion. The extracted data were collected in a spreadsheet (Excel, Microsoft).

A meta-analysis was not possible due to the heterogeneity of the diagnostic tools assessed and the variability in the study designs.

## 3. Results

Initially, a total of 1782 articles was found. The authors considered two additional studies from non-peer-reviewed journals. After elimination of duplicates (*n* = 74), the primary search resulted in 1782 articles. Subsequently, 1749 articles were excluded as non-relevant on the basis of abstract, title and study design. Thirty-three records were screened from the database and another 21 articles were excluded for bias. The full texts of 12 articles were read in order to exclude any additional irrelevant studies. Six articles were included in this review for qualitative analysis. A summary of the clinical studies meeting the inclusion criteria is shown in Table 2 and Table 3. A systematic review [43] was included as noteworthy for the purposes of this article. The search methodology and results are illustrated in the PRISMA flow chart (Figure 1). The conclusions are reported in Table 2 and Table 3.

### 3.1. Description of Included Studies

Six articles [27,29,44,45,46,47] reported the type and the prevalence of malocclusion in pediatric patients affected by AR.

The rate of patients with AR ranged between 18.7% and 5.6% in studies evaluating malocclusion in pediatric patients with a diagnosis of allergic rhinitis [27,29,44,45]. Only two studies reported the prevalence of malocclusion (MO) in children with diagnosis of allergic rhinitis (AR+) and MO in children without allergic rhinitis (AR−) [29,45]. The prevalence of crossbite described in two studies [29,45] ranged from 3.10% to 26%. The prevalence of open bite in children with AR was specified in two of the studies [29,45], and range from as high as 52.3% to as low as 26%.

Only two studies evaluated rhinitis in pediatric patients with a diagnosis of MO [45,46], with one study [46] reporting a diagnosis of rhinitis in 76.4% of patients, with no significant differences regarding the median age (144 months). Two studies [45,46] described the prevalence of crossbite in children with AR, which was estimated to be between 82.8% and 28%, and the prevalence of open bite in children with AR, which ranged between 82.8% and 56%.

Three studies [29,45,46] reported that patients with rhinitis were subjected to the skin prick test (SPT) to identify the etiology. The skin prick technique was used with the standard battery of aeroallergens (*Dermatophagoides pteronyssinus*, *Blomia tropicalis*, *Dermatophagoides farina*, fungal mix, *Blattella germanica* pollen mix, cat epithelium, dog epithelium, histamine (1 mg/mL) and negative control ^®^—FDA Allergen).

The prevalence of dental and ENT findings in the deciduous, mixed and permanent dentitions was described in two studies [27,29].

Only one article [45] determined the association between allergic rhinitis, bottle feeding during the first year of life, non-nutritive sucking habits, and malocclusion in a cohort of children.

Only one article [44] reported cephalometric evaluation in children with allergic rhinitis and mouth breathing.

### 3.2. Quality Assessment and Risk of Bias for Clinical Studies

The risk of bias (R.O.B.) assessment for the included studies is shown in Figure 2. The quality of the included studies was moderate.

In the study by Imbaud et al. [46], the patients were randomly selected in a Hospital Department in São Paulo; the selection criteria might have included selection bias, the randomization might not have been sufficient, there may have been third factor bias due to the pollution in a high-density population center, and it is unknown whether the allergic tests were objective or the classification of malocclusion.

In the study by de Freitas et al. [27], patients of the study group that tested positive for the allergic test were selected from a pediatric hospital; the control group was recruited from a school, and the study group might have introduced selection bias, since it was not randomized. The quality and the standards of the allergy tests are not described, while the use of a compass and the reliability of the intra-operator test might reduce the risk of bias in terms of malocclusion evaluation.

In the study by Souki et al. [29], the selection of the patients took place in an otorhinolaryngology department, which might include selection bias due to the nonrandomized design of the selection; furthermore, there was no blinding for the otorhinolaryngologists. The diagnostic methods for allergic rhinitis are not described, which might have led to nonobjective diagnostic criteria.

In the study by Agostinho et al. [44], risk of bias is represented by the choice to exclude one group from teleradiography for cephalometric analysis; therefore, the tests were not objective, and the allergic tests were defined in a generic way.

In the study by Luzzi et al. [47], the patients were not randomly selected, the questionnaire provided a higher risk of bias in terms of providing knowledge of the aim of the study, and therefore false positives might have been observed in the patients.

In the study by Vázquez-Nava et al. [45], questionnaires were given to the parents of young children to assess allergic rhinitis, and then a skin prick test was performed, but the authors do not refer to a blinding method or randomization tables. Orthodontic examination is also described as being led by pediatric dentists without the use of radiological examinations.

## 4. Discussion

Genetic and environmental factors may cause altered dentofacial growth. Mouth breathing in pediatric patients caused by allergic rhinitis is a common symptom and can lead to several facial skeletal changes, as well as the development of malocclusions [24,29,48,49,50,51,52,53,54,55].

Kawashima et al. [56] and Kerr et al. [57] found that, even in pre-school-aged children (3 to 6 years), the mandible may be retrognathic and posteriorly inclined, especially when the degree of respiratory obstruction rages from moderate to severe. This condition may determine an increase in anterior facial height due to clockwise mandible displacement, resulting in a vertical growth pattern and an open bite tendency [44].

Excess or deficiency in the vertical dimension can be assessed using the facial height index [58].

An increased anterior facial height in relation to the posterior one may indicate a posterior rotation of the lower jaw and a tendency to develop an open bite [29,47,58,59]. The authors found a statistically meaningful association between posterior crossbite and increased overjet. These findings confirm that the effects of oral breathing on craniofacial structures, mediated by the low tongue position and by the modifications of the muscular balance, are more pronounced at the maxillary level, in both transversal and sagittal planes, in children affected by AR [47]. Furthermore, the proportions between the posterior and anterior facial heights were statistically lower in mouth breathing children, indicating a proportionally smaller posterior facial height, and higher lower anterior facial height, confirming that mouth breathing children presented clockwise rotation of the mandible, stimulating greater vertical growth of the anterior region of the face with respect to the posterior region [46,58]. Tourné and Chambi et al. state that mouth breathing should be considered the principal etiological factor of induced augmented vertical growth [60,61].

By contrast, Smith et al. and Warren et al. affirm that it is difficult to determine whether the elongated face is a cause or an effect of augmented nasal air resistance [62,63].

In turn, Vig et al. [51] and Fields et al. [64] state that the association between nasal obstruction and facial growth in children appears to be multifactorial in nature.

Klein et al. [65] and Shanker et al. [66] found no evidence of an influence of mouth respiration on the development of high angle faces, which are the result of different neuromuscular adaptations related to the predetermined genetic pattern.

Shintani et al. [67] concluded that the abnormal facial morphology observed in mouth breathing children may be influenced by genetic and environmental factors (upper airway obstruction).

Trotman et al. [68] found clockwise rotation of the mandible and lower posterior height of the reduced face in children between three and thirteen years with hypertrophy of the pharyngeal and palatine tonsils.

In a study that assessed differences between the sexes, Kawashima et al. [69] found that boys with respiratory disorder during sleep had a higher lower anterior facial height than girls.

Despite this, Vig et al. [51] revealed a significantly higher percentage of nasal breathing in girls than in boys.

On the other hand, those authors found no evidence suggesting that the gonial angle (ArGo.GoMe) presents statistically different values between mouth and nasal breathers. This result is not in agreement with the study by Ahlqvist-Rastad et al. [70], who found increased gonial angles in mouth breathing patients when compared to nasal breathers. Differences in results may be due to the samples used in both studies, especially with regard to the ages of the children evaluated by Ahlqvist-Rastad et al., whose interval was very wide, ranging from 1 year and a half to 14 years of age [70]. In addition, the study was performed in a larger sample, consisting of 122 children, despite the heterogeneous age group. This fact may mask the results, since it involves children in different phases of facial growth. It is important to consider that individuals at 14 years of age have already reached definitive facial dimensions, whereas up until the age of 10, the children have not yet undergone the pubertal growth spurt, and therefore they may still present considerable change in facial morphology. According to Defabianis et al., at around 12 years of age, the upper and lower jaw increase significantly in size, such that 90% of deformities are established before this period [71].

By considering the transverse palatal dimension, in a case–control study conducted on 192 pediatric patients, de Freitas et al. did not find a significant variation between the case and control group, thus confirming that the main influence of alteration in the breathing pattern from nasal to mouth takes place on the vertical plane [27].

Although there is no unanimity, the data seem to suggest a correlation between respiratory impairment and dentofacial deformity. Despite the lack of a complete understanding, maintaining or re-establishing nasal breathing is an important factor for adequate dentofacial growth and development, as demonstrated by Maspero et al. [72].

In a study by Occasi et al. [43], no true relation was confirmed between malocclusion and rhinitis, and vice versa (the rates of association between the two conditions were 29.5% and 38.2%, respectively). The authors concluded that MO and AR were probably concomitant disorders.

The highest risks of bias are represented by the selection criteria, the lack of power of the study, and the sample size, but all of them have been solved by data calculated in preliminary studies. The selection typically occurs in hospitals, usually in orthodontic departments and less frequently in allergic departments; a randomized population study had not been performed; and the period of recruitment was also not considered, even though many allergies are seasonal. None of the included studies adhered to standardized protocol guidelines such as STROBE and CONSORT, which affects the possibility of clearly superposing the results among the authors. Furthermore, third factor biases could strongly affect patients with malocclusion and rhinitis, as genetics could favor less development of the middle third of the facial structures and thus be directly related to both; only a small number of studies consider some of these factors.

## 5. Conclusions

Among all the studies no agreement is reached. In addition, no true relation between maloccusion and rhintis was demonstrated.

Further research and protocols are needed to really evaluate the correlation between allergic rhinitis and malocclusion in the pediatric population.

## Figures and Tables

**Figure 1 children-07-00260-f001:**
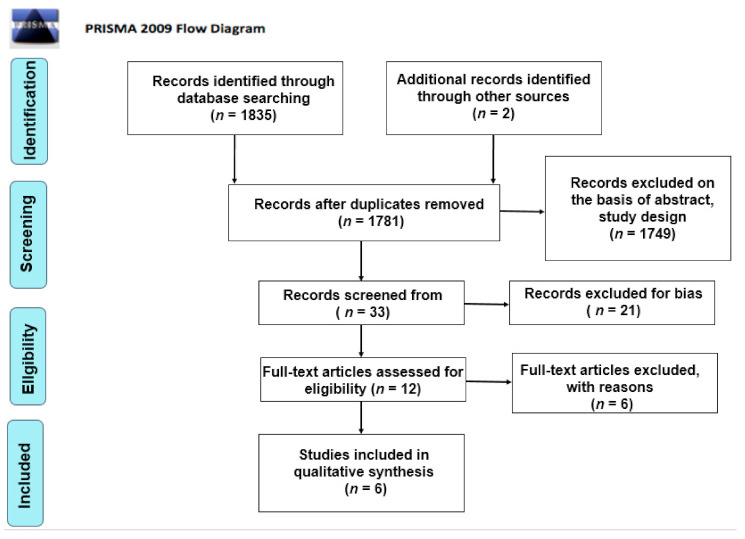
PRISMA flow chart.

**Figure 2 children-07-00260-f002:**
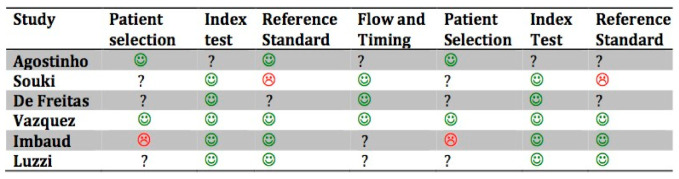
Quality assessment of included studies based on QUADAS-2. The figures in green indicate a low risk of bias: The figures in red indicate a high risk of bias. The question marks indicate that the risk of bias in unclear.

**Table 1 children-07-00260-t001:** Inclusion and exclusion criteria.

Inclusion Criteria	Exclusion Criteria
Articles with reliability or accuracy of the diagnostic method	Articles with lack of methodology of diagnostic method
Information about etiology, diagnosis and therapy during pediatric ages	Articles without information about etiology, diagnosis and therapy during pediatric ages
Articles regarding the relationship between allergic rhinitis and malocclusion	Articles referring to non-pediatric patients (>18 years)
Only articles referring to Humans	Articles referring to non-human subjects
Observational studies	Commentaries
Longitudinal studies	Letters to editor
Prospective studies	Short communications
Case–control studies	Articles referring to other types of rhinitis are not considered such as: non-allergic, infectious, medicamentosa, atrophic, sicca, polypous.
Systematic reviews	
Articles in other languages than English	

**Table 2 children-07-00260-t002:** Characteristics of the studies considering malocclusion (MO) in pediatric patients with diagnosis of allergic rhinitis (AR). AR+ indicates children with diagnosis of allergic rhinitis. AR- indicates children without allergic rhinitis.

Article	Year	Country	Sample (*n*)	Age	Gender M/F(*n*)	Number of patients with AR (%)	Number of patients with non AR (%)	MO in non AR (%)	MO in AR+ (%)	MO in AR- (%)	Crossbite in patients with AR (%)	Openbite in patients with AR (%)	Conclusions
Agostinho [44]	2015	Portugal	70	5–14	41/92	50	/	/	/	/	/	/	Children with AR and Mbreat have longer faces, shorter maxillas and mandibles and a narrowed pharyngeal airway space. No statistical differences between the groups in sagital relationships or in dental inclinations
Souki [29]	2009	Brazil	401	2–12	/	18.7	81.3	/	29 IIcl. 10 IIIcl.	22 IIcl. 7 IIIcl.	26	26	No association between class II malocclusion, anterior open bite and posterior crossbite in patients with adenoids/tonsils hyperplasia or rhinitis
De Freitas [27]	2001	Brazil	192	2–12	/	52.6	/	/	/	/	/	/	No variation of the trasverse palatal dimension. Main alterations on the vertical plane.
Vazquez [45]	2006	Mexico	1160	4–5	582/575	28.8	/	/	43	60	3.10	52.3	AR is a significant risk factor for the development of anterior open bite in primary dentition

**Table 3 children-07-00260-t003:** Clinical characteristics of the studies evaluating rhinitis in pediatric patients with diagnosis of malocclusion. AR+ indicates children with diagnosis of allergic rhinitis. AR- indicates children without allergic rhinitis.

Article	Year	Country	Sample (*n*)	Age	Gender M/F (*n*)	Number of patients with MO (%)	AR+ in patients with MO (%)	AR- in patients with MO (%)	Non-AR in patients with MO (%)	Crossbite in patients with AR (%)	Openbite in patients with AR (%)	Conclusions
Imbaud [46]	2015	Brazil	89	8–15	/	100	62.9	23.6	76.4	82.8	82.8	Increased frequency of AR in children with MO.
Luzzi [47]	2013	Italy	275	5–9	126/149	45.4	59.2	16	/	28	56	Increased risk to develop posterior crossbite and augmented overiet. No association between AR and open.

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
