# Peer review of "Correlation between Malocclusion and Allergic Rhinitis in Pediatric Patients: A Systematic Review"

_children, 2020, doi:10.3390/children7120260_

Round 1

Reviewer 1 Report

It is an interesting paper about the Malocclusion and Allergic Rhinitis in Pediatric Patients.

The  manuscript does meet review criteria.  

I have major concerns :

One of the most concerning issues with the manuscript is that there is no  clearly written what type of allergic rhinitis (AR) presented children.

Perennial AR and seasonal AR are two different  diseases with different number of days of clinical symptoms. There is a lack of research into duration of AR and whether it was treated.

The selection of literature with studies clearly defining the type and duration of AR could help to draw conclusions.

Minor

28 -   Instead of patology – disease

180 - Instead of ediatric – pediatric

Table 2,3 – please explain AR, MO

Please be consistent: pediatric, paediatric

The paper should be reviewed by an English speaking person.

Author Response

The Authors thank very much the reviewer for the precious suggestions. 

We attach the response to their comments.

Kind regards

Reviewer 2 Report

In this manuscript, Farronato et al. performed a systematic literature revision including studies on children and adolescents assessing the prevalence of rhinitis in children with malocclusion and the prevalence of malocclusion in children with rhinitis this systematic review aimed to assess if there is evidence about a correlation between malocclusion and allergic rhinitis in children. The authors fail to establish such correlation, mainly due to high risk of bias of the included studies.

The combination of Embase, MEDLINE, Web of Science Core Collection, and Google Scholar performs best in biomedical systematic reviews, achieving an overall recall higher than 95%. However, I consider the use of Medline, Pubmed, Embase, Cochrane Library, and Google Scholar a very thorough literature review, and would like to congratulate the authors for their effort.

Major comments

  1. The authors provide two search expressions:

A - ("rhinitis, allergic"[MeSH Terms] OR ("rhinitis"[All Fields] AND "allergic"[All Fields]) OR "allergic rhinitis"[All Fields] OR ("allergic"[All Fields] AND "rhinitis"[All Fields])) AND ("malocclusion"[MeSH Terms] OR "malocclusion"[All Fields]) ("rhinitis"[MeSH Terms] OR "rhinitis"[All Fields]) AND ("malocclusion"[MeSH Terms] OR "malocclusion"[All Fields])

 B - ("malocclusion"[MeSH 76 Terms] OR "malocclusion"[All Fields]) AND respiratory[All Fields] AND ("hypersensitivity"[MeSH Terms] OR "hypersensitivity"[All Fields] OR "allergies"[All Fields])

As the syntax of search strategies is database specific, it should be clearly stated to which of the mentioned databases these search expressions refer. Further, in order not to miss any publications related to allergic rhinitis, I would have included other keywords, such as “hay fever” or “pollinosis”.

  1. The authors mention, “no time limits in the search strategy have been considered”; from this statement I take that all studies from the inception of the database were considered, however, the authors fail to mention when was/were the searches performed so that we can have a terminus date.
  2. In Table 2 “Clinical characteristic of the studies evaluation malocclusion in paediatric patients with a diagnosis of allergic rhinitis”, the ninth and tenth columns data are not clear. What is the meaning of “29 II cl” and similar data?
  3. It would be useful to include the sample size in Table 4.
  4. Line 275: “Also third factors bias are strongly affecting patients with malocclusion and rhinitis as genetic could favour less development of the middle third of the facial structures and be directly related to both, only a few studies consider some of them.” This sentence is not very clear to me. What are you implying?
  5. English should be revised.
  6. Mention of the studies and bibliography through the manuscript is not uniform. My preferred form would be “XXXX et al.”

Minor comments:

  1. The PRISMA Flow chart should state the reasons for the exclusion of the studies.
  2. Line 162 should read “São Paulo”.
  3. Line 185 should read “pediatric patients”.
  4. Line 254 should read “rhinitis”
  5. “Pediatric” is written both in American English and in British English spelling. Please correct.

Author Response

The Authors thank the reviewer for the precious suggestions and attach the responses.

Kind regards

Round 2

Reviewer 1 Report

I accept the manuscript, I have no comments.

Author Response

Dear Reviewer

thank you very much for your suggestions to improve the manuscript. We made all of them and now we hope it is ok for you.

We have attached the file with all the responses.

Kind regards
